# Regulation and Consumer Interest in an Antioxidant-Enriched Ham Associated with Reduced Colorectal Cancer Risks

**DOI:** 10.3390/nu13051542

**Published:** 2021-05-03

**Authors:** Stéphan Marette, Françoise Guéraud, Fabrice Pierre

**Affiliations:** 1University Paris-Saclay, INRAE, AgroParisTech, UMR Economie Publique, 78850 Grignon, France; 2Toxalim (Research Centre in Food Toxicology), Toulouse University, INRAE UMR 1331, ENVT, INP-Purpan, UPS, 31300 Toulouse, France; francoise.gueraud@inrae.fr (F.G.); fabrice.pierre@inrae.fr (F.P.)

**Keywords:** experimental economics, meat consumption, consumers’ preferences, risks, colorectal cancer

## Abstract

An economic experiment was conducted in France in 2020 to evaluate consumer attitudes toward two ham products associated with different colorectal cancer risks. We focused specifically on comparing a conventional ham and a new hypothetical antioxidant-enriched ham with a reduced risk of provoking colorectal cancer. Study participants were given descriptions of the two hams before carrying out successive rounds of willingness-to-pay (WTP) assessments. The results show that WTP was higher for the antioxidant-enriched ham than for the conventional ham. WTP estimates were also impacted by providing additional information about the reduction in colorectal cancer risk associated with the antioxidant-enriched ham. Based on the participants’ WTP, we came up with ex ante estimates for the social impacts of introducing the antioxidant-enriched ham onto the market, and we suggest that it would be socially optimal to promote the product. Competition arising from pre-existing product labelling and marketing assertions could greatly limit the market potential of antioxidant-enriched ham, which suggests that alternative approaches may be necessary, such as regulations mandating antioxidant enrichment. These results also concern all countries with high levels of meat consumption.

## 1. Introduction

Colorectal cancer is the third most common type of cancer and the fourth most common cause of death from cancer worldwide [1]. In 2007, 2011, and 2018, the World Cancer Research Fund (WCRF) and American Institute for Cancer Research (AICR) stated that there is strong evidence that consuming processed meats increases the risk of colorectal cancer [2]. Moreover, the World Health Organization (WHO) recently classified processed meat as “carcinogenic for humans” [3,4,5].

Recent scientific studies have found more evidence supporting the positive association between meat consumption and the risk of colorectal cancer. Different hypotheses have been explored to better understand this relationship. Notably, research has examined the effect of heterocyclic amines, *N*-nitroso compounds, and the ability of heme iron content to catalyze the oxidation of dietary lipids and the nitrosylation [6]. Previous work using two animal models of colon carcinogenesis found that nutritional doses of heme iron promote the development of colon cancer [7]. This effect of heme iron has been confirmed in rodent models using purified molecules (hemoglobin or hemin) and both lyophilized and freshly processed meat. Indeed, when given a low-calcium diet of freeze-dried cooked ham and fresh hot dogs, carcinogen-initiated rats displayed a significant increase in their number of mucin-depleted foci (MDF), colonic preneoplastic lesions that have also been observed in humans [8,9].

In another experiment using rats, the consumption of an experimentally produced cured pork product, similar to a cured cooked shoulder ham, promoted carcinogenesis in rats [10]. In humans and in rodent models, the consumption of cured meat increased endogenous nitrosylation, fat peroxidation, and fecal water toxicity, and the association between cured meat consumption and the risk of colon carcinogenesis was specifically linked to luminal lipid peroxidation and the luminal formation of a specific type of *N*-nitroso compound: nitrosyl iron [11].

It has thus been hypothesized that limiting these two endogenous reactions—peroxidation and nitrosylation—could help reduce the risk of cancer associated with the consumption of processed meats [12]. Research has therefore focused on different strategies for changing the food’s physiological effects (chelating heme iron to limit its catalytic capacity and adding antioxidants to directly limit the two reactions), its representation in consumer diets, and the ways in which it is prepared. Some studies have found that adding calcium to the diet or directly adding antioxidants [9] or polyphenols to processed meats during the preparation process sufficiently inhibited the two reactions [10]. These findings therefore serve as a proof of concept that reformulating processed meats can help lower colorectal cancer risk associated with their consumption.

In tandem, dietary antioxidant supplementation in pigs has been found to effectively increase antioxidant status and decrease meat oxidation levels [13,14,15]. Whether this effect in meat products translates into limiting the two endogenous reactions during meat digestion has remained an open question. However, one study explored the ability of dietary vitamin E supplementation in pigs to prevent the formation of carcinogens (nitrosamines and aldehydes) in the resulting processed meat [16]. It also used in vitro enzymatic digestion to mimic the acidic conditions in the human stomach. Pig dietary vitamin E supplementation was found to reduce the formation of aldehydes in processed meat, and tended to reduce the formation of nitrosamines after enzymatic digestion [16]. If these results can be confirmed in an independent study, they suggest that antioxidant supplementation during livestock rearing could represent another technique for lowering colorectal cancer risk.

Indeed, reformulating cured meats to contain protective additives helps diminish the risk of developing colorectal cancer [17]. In other words, this approach could serve as an alternative to providing nutritional recommendations or warning directly to consumers. Despite their limited efficiency for changing purchasing behaviors, public recommendations broadcast by national and international agencies are often favored for avoiding [2] or limiting [18] the consumption of red and processed meats. If adopted by consumers, recommendations to avoid or limit the intake of processed meat could reduce the colorectal cancer burden [19]. Consuming greater quantities of legumes, fruits, and vegetables also increases antioxidant intake and reduces the risk of colorectal cancer, the latter largely being linked to the heme content of processed meat [20]. However, many people do not pay attention to nutritional messages [21]. In particular, individuals of lower socioeconomic status and with a higher intake of processed meat are less receptive to nutritional messages [22]. Indeed, consumer desire to reduce meat consumption is often thwarted by dietary habits and cultural factors, which favor regular meat consumption [23]. The existence of dietary routines helps explain the overconsumption of animal-based proteins in many developed countries, while the health benefits of alternative diets are overlooked by many consumers [24,25].

Given this context, it is particularly worthwhile to study the alternative approach consisting of reformulating processed meats to promote healthier outcomes [26]. For example, processed meat products could be enriched with natural antioxidants, which significantly reduces the risk of colorectal cancer associated with processed meat consumption [10].

Our study examines consumer attitudes, in the form of willingness-to-pay (WTP) estimates, to a potential product to be sold in France. Our hypothetical product was a type of cooked ham, because it is the most popular processed pork product, representing 30.4% of the category’s market share in France [27]. Since this new antioxidant-enriched ham was neither authorized for sale nor sold in France at the time of the experiment (February 2020), we estimated the hypothetical WTP, which means that participants expressed their preferences for potential products without having to purchase anything at the end of the experimental session. We focused on the influence of informational messages on study participants’ WTP for different types of hams. WTP can reliably indicate future demand for a new product that is not yet on the market [28]. The main contributions of this research stem from (1) its examination of consumer preferences for a new potential product with pronounced health benefits and (2) its ex ante WTP estimates that point toward the use of regulatory approaches. We now turn to the presentation of this experiment.

## 2. Material and Method

This section successively details the experiment and the methodology used for the econometric analysis of WTP.

### 2.1. The Experiment

#### 2.1.1. The Sample

We conducted the experiment in Dijon, Burgundy in France, in multiple sessions on 3 and 4 February 2020. This period came before the beginning of the COVID-19 crisis in France, which started at the end of February with the first detected cases [29]. Participants were recruited through the Chemosens PanelSens database built by the Centre des Sciences du Goût et de l’Alimentation, INRAE, Dijon.

A sample of 146 participants was randomly selected based on the quota method, and for being representative of socioeconomic status for the population of the city. For age, we targeted two specific groups of participants, taking into account the fact that the risk of colorectal cancer is significantly higher above the age of 50 [2]. Thus, we recruited 50% of participants younger than 50 and 50% of participants older than 50 with a relatively high risk of colorectal cancer.

During the recruitment, participants were contacted by email. They were informed that the experimental session would focus on pork consumption and it would last about one hour with a €10 participation fee. Only participants who ate pork or processed meat, even occasionally, were selected. On 3 and 4 February 2020, each experimental session lasted 50 min on average, and included between 14 and 18 participants who attended one session.

The rest of the paper considers replies and WTP by 139 participants, since seven participants were withdrawn from the analysis because of inconsistent replies regarding WTP with the multiple price list (see below). Table 1 describes socio-demographic variables collected in the exit questionnaires at the end of the sessions. Recall that the distribution of ages was voluntarily distorted by having half of the participants younger than 50 and half of the participants older than 50, since the risk of colorectal is higher for people older than 50. Note that the levels of education for the 139 participants are relatively representative of the French population.

#### 2.1.2. The Products

Because (i) the new enriched ham was not sold at the time of the experiment and (ii) we could not fully guarantee the freshness of the products, in particular from the lab to the participants’ fridge, no products were sold at the end of the experiment. In the absence of products given at the end of the experiment, we elicited the hypothetical WTP. Despite the risk of hypothetical and upward biases of WTP, the lab is a practical place for eliciting well-informed, thoughtful preferences with a tight control of the revealed information. Even if the hypothetical WTP is likely to be upward biased, recent contribution seems to downplay the risks of biases for private goods. By comparing hypothetical and non-hypothetical responses, it was shown that the marginal WTP for a change in quality/characteristic is, in general, not statistically different across hypothetical and real payment settings [30].

With one picture of ham shown on one paper sheet, participants indicated the choices they would make in supermarkets (see Appendix A). The picture of one product sold by a popular brand was taken into account. The brand was hidden to create a generic product. Many hams are widely produced and sold under different brands, including supermarket brands. The picture presented a 160 g package containing four slices of ham. This conventional product with four slices was selected because of its vast presence in supermarkets in Dijon and also in France.

#### 2.1.3. Timeline of the Experiment

At the beginning of the experiment, some initial explanations were read, and participants signed a consent form. We insisted on the fact that all of their replies were anonymous, since participants were identified by a number. We mentioned the fact that no product would be sold or given at the end of the experiment. We asked participants to indicate choices as if they were in a supermarket. We insisted on the absence of “good” or “bad” replies, but rather on the possibility to freely indicate choices reflecting their preferences.

Six rounds of WTP elicitations were organized with successive messages and were revealed with the notation #*R* for a round *R* = {1,2,3,4,5,6}. Round #1 was realized with only a few explanations about the weight and the ham packet with fours slices. This ham represented in the picture and sold in France at the time of the experiment is named “conventional” for the rest of this paper. A few explanations were also given about the multiple price list (see Appendix A the forthcoming subsection). Participants filled out this price list for the conventional ham. After this initial round #1, and at each round #*R* with *R* = {2,3,4,5,6}, one message was given to participants on a paper sheet and read by the organizer. After each message, each participant successively filled in multiple price list(s) for one or two hams.

At round #2 and #3, only the conventional ham was considered. Two successive messages about the link between colorectal cancer and processed meat at round #2 and about consumption advice for limiting the risks of colorectal cancer at round #3 were revealed to participants (messages are presented in the next subsection).

At round #4, the new enriched ham was introduced, and participants were asked their purchase intentions for the two products, the conventional ham on one page and the new enriched ham on another page. At round #4, this introduction of the new ham enriched with antioxidants was accompanied with an explanatory message about its potential for reducing the risks of cancer. At round #5, a message about the effects of different antioxidants on the taste and color of the new product were revealed to participants. Participants were divided in three subgroups with different antioxidants and a few differences regarding their effect on the product (see the next subsection). At round #6, we revealed a last message about the impact of the new ham enriched with antioxidants, providing scientific information about the mechanisms allowing the reduction of the risks of cancer. 

At the end of a session, after the six rounds, participants filled in an exit questionnaire with 27 questions (see Appendix B) and received the €10 indemnity. 

#### 2.1.4. The Messages about the Products

At rounds #2 to #6, different types of messages about the products were communicated to participants before WTP elicitations and quantity choices. The messages were written after studying articles coming from the toxicological field. The messages were relatively short, because previous works underline that a short message is more efficient than a long message with complex information [31]. The order of the messages did not vary, since they followed a progression towards the introduction of the new enriched ham and its impact. For the message of round #5, participants were divided into three subgroups, with 44 participants for the subgroup G1, 45 participants for the subgroup G2, and 50 participants for the subgroup G3.

The five messages translated from French and preceding the WTP determinations were the following.

##### Message at Round #2 with Generic Explanations about Colorectal Cancer

Recent scientific evidence suggests that excessive consumption of red meat and processed meats is a significant risk factor for the development of colorectal cancer (affecting the colon and rectum). Thus, the French National Program for Health and Nutrition (“Programme National Nutrition Santé (PNNS)” in French) recommends limiting the consumption of cured meat (charcuterie) to 150 g per week.

Epidemiological studies have shown that people who consume more than 50 g/day of processed meats see the risk of colorectal cancer (concerning the colon and rectum) increase by 16%. In addition, according to the ICRC (International Center for Research on Cancer), 10% of colorectal cancers in France come from the excessive consumption of processed meats. Colorectal cancer is the third most common cancer in France.

However, a reasonable consumption of meat and processed meats brings nutritional benefits, with the provision of iron (namely, heme iron that is very absorbable and important to fight anemia), vitamin B12 (essential for neurological functions, with meat products as the main source), and proteins of good nutritional quality.

##### Message at Round #3 with Consumption Advice for Limiting the Risks of Colorectal Cancer

In order to avoid the harmful effects of excessive consumption of processed meats, in particular regarding the risk of developing colorectal cancer, it is advised to:−limit the consumption of red meats to 500 g per week and the consumption of processed meats to 150 g per week, and, among the processed meat, to favor white ham and poultry ham;−favor during meals the consumption of natural foods rich in antioxidants and fibers, such as fruits and vegetables, to accompany red meat or processed meats.

##### Introduction of the Enriched Ham at Round #4 and the Message about This New Ham Enriched with Antioxidants

Work by the INRAE (formerly INRA) has shown that antioxidants protect against the risk of colorectal cancer associated with the consumption of processed meats. Consequently, the INRAE has decided to develop a research program aimed at developing new charcuterie/processed products that limit the risks of colorectal cancer by working in collaboration with breeders and manufacturers of the charcuterie/processed meat sector.

In this context, these new antioxidant-enriched products are obtained by adding natural antioxidant elements to processed meat during its production for G1 and G2 or pig feed during breeding for G3. These new enriched products could soon be introduced on the French market.

##### Message at Round #5 on the Effects of Different Antioxidants on Taste and Color

For subgroups G1 and G3:

The antioxidants added to processed meats during their production for G1 or pig feed during breeding for G3 are from the tocopherol family, such as vitamin E. They are natural additives, and they do not alter the taste of enriched ham compared to conventional ham.

The presence of these tocopherols does not change the color of the ham.

For subgroup G2:

The antioxidants added to processed meats during their production are from the polyphenol family. They are natural additives, and they do not alter the taste of enriched ham compared to conventional ham.

The presence of these polyphenols slightly changes the color of the ham, which becomes slightly more purple compared to conventional ham due to the addition of polyphenols (antioxidants) extracted from red wine or fruit (pomegranate, for example).

##### Scientific Message at Round #6 about the Impact of the New Enriched Ham

Scientific evidence of the biological effectiveness of the modification of cured meats produced with tocopherols added to processed meats during their production for G1, to pig feed during breeding for G3, or polyphenols added to processed meats during their production for G2 has been provided by INRAE research. This evidence was acquired from animal experimentation and a nutritional study in humans consuming processed meats.

The mechanism is as follows: heme iron present in meat products is the main agent responsible for the risk of cancer, even if it also has a nutritional advantage, because it is easily absorbed by the human body. This heme iron causes a lipid oxidation reaction in the diet, favoring precancerous cells in the colon, and a reaction with nitrites to form toxic compounds.

The presence of antioxidants in enriched ham avoids these two reactions (lipid oxidation and the reaction with nitrites), and thus significantly limits the risk of developing colorectal cancer.

#### 2.1.5. Mechanism for Eliciting WTP

A multiple price list (payment card) was used for eliciting the WTP (willingness-to-pay) of each product. At each round, participants chose whether or not they would buy the product for different prices (see Appendix A). These prices, varying from €2.20 to €3.80, were selected because they epitomize a representative range of prices observed in supermarkets at the time of the experiment. Indeed, from our observations in Dijon supermarkets, the market prices ranged from €2.40 to €3.20 for one 160 g package with four slices of ham.

Participants had to fill out 17 lines for each product and for each choice (see Appendix A). For each price, they had to check off either “yes”, “no”, or “maybe” regarding their purchase intents. For each product and for each round of choice *R* with *R* = {1,…,6}, the WTP was determined by taking the highest price linked to the choice “yes” (with the following highest price on the paper sheet implying a reply of “no” or “maybe”). For a participant *i*, the WTP at a round *R* for a ham H is denoted $\;WTPH\#R_{i},$ with *H* = {C,E} for conventional/enriched. When one participant only replied “no” or “maybe” to all lines of the multiple price list, the selected WTP was arbitrarily equal to €0. An alternative configuration could be a value between €0 and €2.10, because it is lower than €2.20, the lowest price of the multiple price list. When one participant only replied “yes” to all lines of the multiple price list, the selected WTP was equal to €3.80, the highest price of the multiple price list. For respondents switching twice at low and high prices, the highest “yes” was recorded as the WTP for the analysis. This paper only considers the WTP with the highest “yes” for the 146 participants. Among them, 17 participants were also characterized by a lower limit, LL < WTP, for which the reply was “maybe” or “no” for prices lower than the LL; Ref. [32] underscores the consequences with multiplicity of market equilibria.

Advantages and drawbacks of multiple price lists were brought forward by some authors [33]. The main advantage of such a list is its simplicity, guaranteeing a direct participant’s understanding. The possibility to check off “maybe” also captures consumers’ hesitation. Conversely, one drawback is the interval response, eliciting interval data rather than point estimates for the WTP. With our experiment, the 10-cent interval guarantees a sufficient degree of precision for the elicited WTP. A second disadvantage is the framing effect, with a psychological bias towards the middle of the multiple price list for choices made by the participants [33]; they controlled for this effect by changing the boundaries of the multiple price list. In this paper, we did not control this framing effect by changing the boundaries, since we focused on the impact of the revelation of successive messages. As all methodologies eliciting WTP, the multiple price list has some limitations, but it is particularly tailored to a protocol insisting on the revelation of new messages.

### 2.2. Method

#### 2.2.1. Analysis of WTP

The software *R* was used for the statistical analysis. WTP differences between the rounds for the same product or between products for a same round were tested two-by-two by using the paired difference test, namely the Wilcoxon test.

Additionally, the impacts of different messages on the WTP were estimated using econometric estimators. We restricted our attention to the last three rounds with the presence of both hams, *H* = {C,E} for conventional or enriched. We pooled the observations corresponding to participants’ WTPs elicited for both hams in the successive rounds *R* = {4,5,6}. Given that each participant *i* wrote three WTPs for each product (leading to six WTPs for the two hams with the pooled WTPs), errors related to these WTPs were potentially correlated to each participant. The random effect imposes constraints on the structure of the variance/covariance matrix. Furthermore, the WTP cannot be negative, and is left-censored at zero, which is why we used the random effects Tobit estimator. Each ham, conventional or enriched, was identified by a dummy variable equal to 1 for a given product, and zero otherwise. The type of information shared in a given round was identified by a dummy variable equal to 1 when it was shared before the WTP elicitation in a given round (and 0 otherwise). Thus, the dummy variable Message on Antioxidants is equal to 1 in round #5, and 0 otherwise. The dummy variable Scientific Message on Health Impact is equal to 1 in round #6, and 0 otherwise.

For a participant *i*, let $WTPH\#R_{i}$ with *H* = {C,E} denote the participant *i*’s WTP as the dependent variable at round *R* with *R* = {4,5,6}. Let *X_R,i_* denote the matrix of explanatory variables indicating the rounds of information with the dummy variables (1/0) presented in the previous paragraph, the socio-demographic and perception variables (from the exit questionnaire). The random model for the WTP can be written as:(1)$$WTPH\#R_{i}=\beta_{1}X_{R,i}+\varepsilon_{R,i}$$
with the vector of coefficients $\beta_{1}$ being estimated and $\varepsilon_{R,i}$ being the error term. The econometric estimations were conducted with two models, model 1 with only the related dummy variables estimated. The impact of socio-demographic variables coming from the exit questionnaire were also tested in model 2, only keeping the variables that had a significant impact. Appendix B presents a brief overview of the exit questionnaire.

#### 2.2.2. Analysis of Regulation and Economic Surplus

The software Excel was used for estimating the impact of regulation on economic surplus. We considered economic surplus for analyzing policy options that could improve products and the consumers’ situation. We examined two configurations, namely the one under perfect information about different types of products and the other one under a mandatory standard, imposing the antioxidant to all products without revealing information. Note that the configuration under perfect information is hypothetical compared to the real context, since consumers have difficulties in knowing and/or recalling information, because of imperfect memory and information overload [34].

We estimated the impact of the emergence of these enriched products under perfect information or a mandatory standard on participants’ surplus by using the observed WTP equal to the highest “yes” and directly coming from the lab (and not the predicted WTP coming from econometric estimations [35]). To convert these WTPs into demand curves, we assumed that each participant would purchase one unit, providing the largest surplus approximated by the difference between the WTP and market price [36,37]. This choice was inferred, because the “real” choice was not observed in the lab experiment, which only elicited the WTP.

These surpluses are estimated ex ante, namely before the introduction of new products. The participants’ surplus with the integration of the effect of ignorance leads to a positive surplus variation only when the purchasing decision changes after the revelation of information, which is fully compatible with the value of information defined under the welfare theory [38]. The considered model is a simplified estimation of the market effects with one or two products, which is a proxy of market adjustments with many products. For simplicity, we abstracted from producers’ profits by concentrating on consumers’ surpluses and the potential new demand. Expressions of surpluses are detailed in Appendix C. We now turn to the results.

## 3. Results

### 3.1. Analysis of WTP

This section focuses on the evolution of WTP at different rounds. Recall that the WTP is determined by taking the highest price linked to the choice “yes” (and zero if not “yes”) in the multiple price list. The impact of revealed messages on the expected WTP for one packet of 160 g is shown in Figure 1. Expectations of the WTP take into account all WTPs, including the non-engaged bidders with the WTP equal to zero, explaining why the averages may be lower than €2.2, the lower bound of the multiple price list. The rounds *R* = {1, …, 6} are represented on the x-axis with indications of the types of information, and the expected WTP in €, denoted E(WTP), are on the y-axis. In Figure 1, the indicator Δ isolates the significant impact of a single round of additional information for the same ham, and/or between the two types of hams for the same round. The curve for the enriched ham only starts at round #4, when it was introduced.

Figure 1 underscores that the messages at rounds #2 and #3 did not change the initial WTP for conventional ham at round #1. As a possible explanation, it should be noted that these first messages mainly described the issue related to cancer and provided consumption/diet advice, while the WTP elicitation only concerned one specific product. At round #4, the introduction of enriched hams significantly decreased the WTP for the conventional ham, while the WTP for the enriched ham was close to the initial WTP for the conventional ham (namely, the one at round #1). At round #5, the differentiated messages about the type of antioxidants did not significantly change the WTP. Eventually, the WTP for conventional and enriched hams were impacted by the last message about the reduction in the risk of colorectal cancer coming from the enriched ham. Indeed, the last message detailing the scientific results coming from the enriched ham had a strong impact on both types of ham at round #6. In other words, this detailed message mainly led to a reduction in the WTP for the conventional products, although the WTP for the new enriched ham significantly increased.

The averages presented in Figure 1 take into account heterogeneous WTPs. Table 2 provides a few statistics about the types of consumers for going beyond the averages presented in Figure 1. Among the participants, we accounted for 11 “non-engaged” participants who bid zero during each round for all products. We also accounted for 21 “indifferent” participants, who did not change their WTPs for any of the products at round #6.

Table 2 shows that only a few participants were reluctant or a boycotter of the new enriched ham, namely only 12 participants. These participants seemed concerned by the presence of new additives, even if these antioxidants were natural. Table 2 clearly shows an interest in antioxidants by many participants. From the last three lines at the bottom of Table 2, we had 95 participants who were characterized by a higher WTP for enriched ham compared to conventional ham after the entire revelation of information, namely with WTPE#6 > WTPC#6. They represented 68.3% of the participants, indicating a substantial majority interested in reducing the risks of having colorectal cancer. This finding suggests that there is a potential market for this new enriched ham.

The impact of differentiated messages on the types of antioxidants at round #5 is now presented. Figure 2 comes from Figure 1, except that we detail the subgroups and restrict our attention to last rounds *R* = {4,5,6} with the presence of both hams. The sensitivity of the participants to the messages explaining the effect of antioxidants at round #5 was limited, since only subgroup G1 exhibited a significant reaction. In subgroup G2, receiving the message that polyphenol may influence the color of the ham, the visible decline in the WTP for the enriched ham at round #5 was not significant, because it came from a few participants with a sharp drop for this WTP.

Next, the econometric estimations based on a random Tobit estimator with dummy variables that represented different messages confirmed the results of Figure 1. The econometric estimation of the WTP is presented in Table 3 to measure the impact of different messages when both hams were presented at round *R* = {4,5,6}. Table 3 presents two regressions that measured the effects of the different rounds of information on the WTP for one packet of ham (see Section 2.2). The coefficient linked to one round measured the impact of this round on the WTP of this product. The omitted variable was for round #4. The first regression only accounted for dummy variables indicating the type of ham and the round. The second regression accounted for dummy variables with one additional socio-demographic variable from the exit questionnaire and identified as statistically significant.

Table 3 confirms the results of Figure 1 and underlines a significant difference in the WTP between both products at round #4, with a coefficient that had a higher value for the enriched product (2.870) than the value for the conventional product (2.713) for model 1. We also observed a significant influence of the scientific message focusing on the impacts of antioxidants on health revealed at round #6. This message significantly and negatively affected the WTP for the conventional product, with a coefficient equal to −0.341 in model 1, increasing the differences between the WTPs of both products. With model 2 (right column), age was the only socio-demographic variable to significantly influence the estimation. The dummy variable (age > 50) equal to 1 for participants older than 50 significantly influenced the WTP for the enriched ham, with a coefficient equal to 0.312. Aged participants, who were the most concerned by the risk of colorectal cancer, were particularly sensitive to this new enriched ham. Alternative analyses not detailed here showed that replacing zero by €2.1 if no “yes” was checked off on the list changed the values of the WTP, but not the nature of the econometric results about the impact of new enriched ham.

Eventually, the previous developments considered the highest “yes” on the price list sheets for determining the WTP, and a WTP equal to zero if no “yes” was checked off. An important fact to notice came from the high proportion of participants checking off one or several “maybe” answers for prices higher than the highest “yes”. Just for the choices at the first round with only the conventional product, 69.7% of the participants replied “maybe” at least once above the highest price, to which they replied “yes.” This high percentage indicates that many participants felt hesitation in the face of these products. For presenting the importance of “maybe”, we took the highest value “maybe” checked off and noted MaybeH, integrating it to the value Max[WTP,MaybeH]. This new value combines the WTP for participants without any “maybe” checked off in the multiple price list and the MaybeH that is the highest “maybe” in this list when MaybeH > WTP.

Starting from Figure 1 with the original curves converted to dashed curves, Figure 3 shows the average values of Max[WTP,MaybeH] for both products (with E() denoting the expectation). The integration of MaybeH led to a significant translation of both curves towards higher values. The relatively high values of MaybeH marking a hesitation suggested that the WTP could increase if advertising and promotion efforts were made to persuade individuals to consume these new products impacting human health. This hesitation was overlooked by previous studies.

To conclude this subsection, it should be noted that the difference in the WTP between both products was relatively high at round #6 after the full revelation of information. We now turn to the study of some regulatory tools that could help market configuration get closer to the situation at round #6.

### 3.2. Analysis of Regulation and Economic Surpluses

At the time of the experiment, no ham enriched with antioxidants was offered to consumers on the French market. Indeed, the market is unlikely to lead to a systematic selection of antioxidants by farmers or firms because of the lack of knowledge by consumers and the weak incentives in the supply chain. We now estimate the consequences of two regulations on participants’ surplus/welfare by using the observed WTP related to the highest “yes” (see Section 2.2). In this section, we abstract from the producers’ profits to focus on the consumers’ surpluses equal to the overall welfare (see Appendix C).

Figure 4 measures the impact of the introduction of enriched ham on the market under perfect information, which is a hypothetical case (see Section 2.2). The perfect information for consumers provides a full incentive to producers to offer enriched ham. As the price of the enriched ham $P_{E}\;$is unknown, different scenarios regarding this price $P_{E}\geq P_{C}$ are considered, starting from the price $P_{C}=€\;2.5$. For each chart, the price of this enriched ham $P_{E}\;$in € is represented on the x-axis. On the chart at the top, the average welfare variation E[W2−W1] (in €) is represented on the y-axis (with precise notations given in Appendix C). On the chart at the bottom, the numbers of participants predicted to purchase the enriched ham or conventional ham for different prices $P_{E}\;$are indicated.

Figure 4 clearly shows a high social benefit linked to the introduction of enriched ham under perfect information. The chart at the top reveals that the average social benefit E[W2−W1] for one purchased unit is relatively high, in particular for values of $P_{E}\;$close to the price *Pc* of the conventional ham, namely for values between €2.5 and €2.9. Indeed, with these prices, many participants are predicted to purchase the enriched ham (based on the WTP elicited at round #6). Note that, conversely, when the price of the enriched ham is very high ($P_{E}\geq €3.8$), there is no demand for the enriched ham, but consumers would benefit from this perfect information, since some of them would stop buying it in accordance with their informed preferences. This high social benefit is reflected by the large number of participants choosing enriched products (in the chart at the bottom), compared to a maximum of 139 participants, in particular for a price $P_{E}$ such that $€2.5\leq P_{E}\leq €2.9$. Figure 4 clearly suggests that it would be socially optimal to promote this new enriched ham. 

### 3.3. Mandatory Standard Imposing the Enrichment

Because of difficulties in credibly informing consumers, an alternative could consist of enforcing a mandatory standard imposing antioxidants in ham, which would change the nature of conventional ham, even if consumers do not know it. By starting from the chart at the top of Figure 4, Figure 5 shows the welfare impact, E[W3−W1], coming from a mandatory standard. Even if the related curve with the standard is lower than the curve under perfect information, E[W2−W1], because of a lack of product diversity hurting consumers reluctant to the enrichment, this standard would also bring a social benefit with a positive curve E[W3−W1]. 

### 3.4. Participants’ Perceptions

The exit questionnaire at the end of each session (see Appendix B) provides indications about participants’ perceptions, complementing the previous results.

Table 4 helps explain the difficulty for the enriched ham to emerge on the market, mainly because of label proliferation, impeding the configuration of Figure 5. Table 4 focuses on questions related to the participants’ perceptions regarding the existing labels and allegations posted on ham packets, sold on the French market at the time of the experiment. Participants’ opinions clearly show that there is a kind of “competition” between these six labels and three allegations existing at the time of the experiment. The Organic and Label Rouge labels are clearly seen as the best ones for guaranteeing quality and guiding the future purchases of participants. Beyond the Label Rouge label, the role of other labels seems weaker, with a low percentage of participants mentioning them. For the most useful, the Nutriscore label also plays a role with Organic and Label Rouge. Allegations show a preference for the absence of antibiotics. Regarding allegations, hams without antibiotics dominate participants’ preferences.

Eventually, from the exit questionnaire at the end of the sessions (see Appendix B), Table 5 focuses on participants’ intentions regarding the potential use of colorectal cancer screening aimed at reducing the risks of cancer [39]. As already explained, this screening test is advised to citizens older than 50 who are the most at risk. As underscored in Table 3, showing a high WTP for enriched ham in dwindling risks, Table 5 confirms the consumers’ interest in reducing the potential risks of colorectal cancers. The percentages of participants in favor of the screening test are very high, marking a sustained sensitivity for questions related to the risks of cancers. The replies by young participants with an age lower than 50 are almost similar to the ones by the older participants with an age older than 50. They confirm the participants’ sensitivity observed in Figure 1 and Table 3 and Table 4.

## 4. Discussion and Policy Implications

### 4.1. Discussion

To the best of our knowledge, this study is the first to examine the WTP for a hypothetical antioxidant-enriched ham, the consumption of which could reduce colorectal cancer risk. The WTP estimates show that participants displayed a strong preference for the potential antioxidant-enriched ham. As this ham would use natural additives, such as tocopherol or polyphenol, it could be easily authorized for sale by regulatory authorities, but the proliferation of labels underlined in Table 4 could explain the difficult emergence of this ham on the French market.

Our results confirm findings from previous studies indicating that participants are willing to pay a positive premium for enriched meats compared to conventional meats (meat enriched with polyunsaturated fatty acids [40]; meat enriched with plant sterols [41]; and elsewhere with [42]). We observed a positive willingness to pay for enriched ham with natural compounds, such as tocopherol and polyphenol. However, previous research had not examined how WTP estimates could inform regulatory approaches.

We found that participants paid attention to the information provided to them. The information provided during round #6 significantly and negatively affected WTP estimates for the conventional ham, but not for the antioxidant-enriched ham (Table 3). This difference can be explained using prospect theory, which states that gains and losses are valued differently [43]. More specifically, utility variation was found to be convex for losses and concave for gains, which means that the shift in the impact was more pronounced for losses than for gains. The design of our experiment, in which both positive and negative information was provided, could be seen as testing predictions of the prospect theory, which posits in this specific context that information about losses should have a greater impact than information about benefits with regards to the antioxidant-enriched ham.

Our results also underscore that, from a societal perspective, it should be possible to successfully introduce healthy hams into the market. Because comprehensive information can lead to an increase in welfare/surplus (Figure 4), recommendations and generic marketing (even if imperfect) could result in consumers making major dietary substitutions, thus improving their health via the consumption of better-quality ham. However, although recommendations and generic marketing could increase consumer knowledge and sensitivity, they are not a panacea for changing consumer behavior. They often fail to modify consumption given conditions of information proliferation and consumers’ imperfect recall of information [44]. The marked potential for social benefits observed in this study (Figure 4) would decrease in contexts in which imperfect information was provided.

Indeed, the results observed in Figure 4 assume that perfect information is provided, an impossible outcome in real life. Several factors are likely to hamper information diffusion, and thus the market success of antioxidant-enriched ham. First, revealing generic knowledge about colorectal cancer risk could create confusion about the health effects of processed meat, damaging the reputation of the entire industry. To deal with this potential issue, it would be crucial to carry out extensive communication, informational campaigns, and generic advertising, which would obviously be very costly. The results also underscore that consumers must evaluate the many labels and marketing assertions on ham packages (Table 4). Competition related to these two factors may greatly limit the market potential of antioxidant-enriched hams signaled with a label. In other words, the proliferation of labels and marketing assertions serves to diminish the potential impact of creating a new label to signal the presence of antioxidants in ham to consumers.

### 4.2. Policy Implications

The challenge associated with providing credible information to consumers is likely to strongly discourage companies from developing such new products. As a result, a possible alternative strategy could be establishing regulations requiring ham to be supplemented with antioxidants, which would change the nature of conventional ham and thus eliminate the need for any evaluation on the part of consumers. Similar regulations already exist. For example, it is mandatory in many countries for ethanol and biodiesel to be added to gasoline for automobiles. We found that requiring ham to be enriched with antioxidants would have a social benefit (E[W3−W1]), which would be greater than €0.1 per package for a price $P_{E}<€2.90$ (see the red curve in Figure 5). However, the results from the exit questionnaire underscore a relative reluctance on the part of participants regarding a mandatory enrichment scenario: only 38.1% agreed with the idea.

A regulation mandating the antioxidant enrichment of ham would obviously impact the supply chain, including farmers, but this topic was not addressed in this study. However, we could speculate that a portion of the social benefits (Figure 5) could be captured by producers if they asked for slightly higher prices. Although we kept prices constant here, we could study this question by extending the model to include endogenous prices and profits. In particular, the producers that we queried told us that supplementing with tocopherol would increase the cost of a 160 g package of ham by an estimated 1.6 cents (€0.016), which seems affordable if producers can benefit from a portion of the social benefits (Figure 5) via higher prices. If this cost of 1.6 cents is fully passed onto consumers and added to the initial price equal to €2.5 of the conventional ham, the new price of the enriched ham would be equal to $P_{E}=€2.516$, leading to a welfare estimate of E[W3−W1]=€0.19 per package, which should encourage reflection about employing a regulatory strategy.

Eventually, we extrapolate this average surplus variation for one package, equal to E[W3−W1]=€0.19, to the overall consumption of pork in France over one year. In 2018, the average pork consumption per inhabitant in France was equal to 33 kg/year. Because the surplus variation E[W3−W1]=€0.19 is linked to one pork package of 160 g, the variation for 1 kg is given by (1/0.16) × (€0.19). We consider this variation for the 65 × 10^6^ inhabitants in France. Thus, the overall variation over a year for the whole population is calculated by multiplying E[W3−W1]=(€0.19) to 33 × (1/0.16) × 65 × 10^6^, and is equal to €2.547 billion. This important surplus variation reflects the high benefit of intervening and offering new products. However, such an extrapolation is limited, since no price adjustments of the products is considered, and since no quality differentiation regarding hogs is taken into account. Alternatively, as an extension, it would be possible to determine the effects of a per-unit tax on conventional ham and a per-unit subsidy for antioxidant-enriched ham on WTP estimates with a view of changing eating habits [45,46].

Even if the lab experiment only considers 139 French participants, these important surplus variations lead to important regulatory implications beyond France, in particular for other countries with high levels of meat consumption, such as Argentina, Germany, or the United States. The question of developing new enriched meat should also be considered and studied for these countries.

## 5. Conclusions

Although laboratory studies have their limitations, our results underscore the existence of social benefits and consumer preferences for antioxidant-enriched ham that could reduce colorectal cancer risk. Furthermore, they suggest that regulations mandating the antioxidant enrichment of ham should be seriously considered. This debate can be extended to beef and lamb, with possibilities of adding tocopherol during their feeding at the farm. Beyond France, these results also concern many other countries with high levels of meat consumption per inhabitant.

Overall, these findings could help inform debates around the best ways to improve consumer health and nutrition. Our study took the innovative approach of using an ex ante (rather than an ex post) method for estimating the possible impacts of a new potential product and of regulations that could simultaneously maximize market allocation and consumer health.

## Figures and Tables

**Figure 1 nutrients-13-01542-f001:**
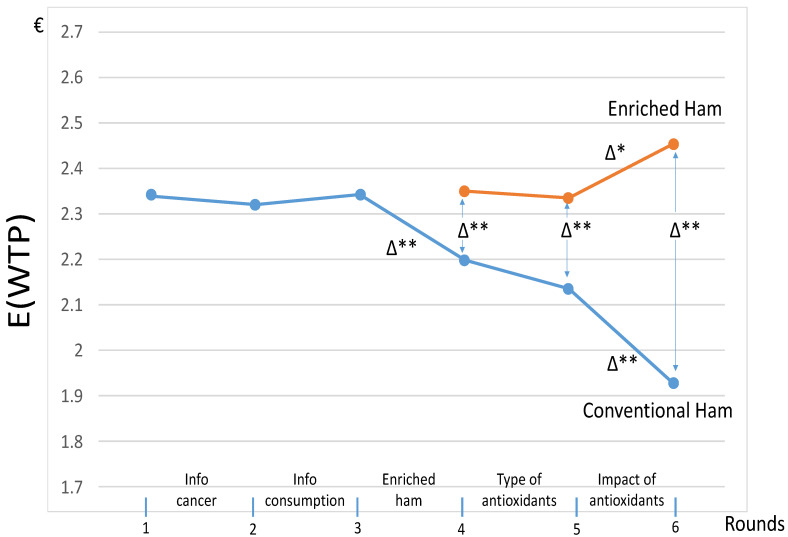
The average WTP for one packet of ham (€) and the influence of the messages. Note: 139 participants were taken into account. Δ* denotes a significant difference at 5%, and Δ** denotes a significant difference at 1%, as tested by the Wilcoxon test for comparing paired sample choices. For each product with Δ indicated above the corresponding line, we tested for significant differences in the WTPs elicited between two successive rounds *R*, with *R* = {1, …, 6} for the conventional ham and *R* = {4,5,6} for the enriched ham. For the same round with Δ indicated between the points of different products, we tested significant differences between WTPC#4 and WTPE#4, WTPC#5 and WTPE#5, and WTPC#6 and WTPE#6 for the different rounds and for *R* = {4,5,6}.

**Figure 2 nutrients-13-01542-f002:**
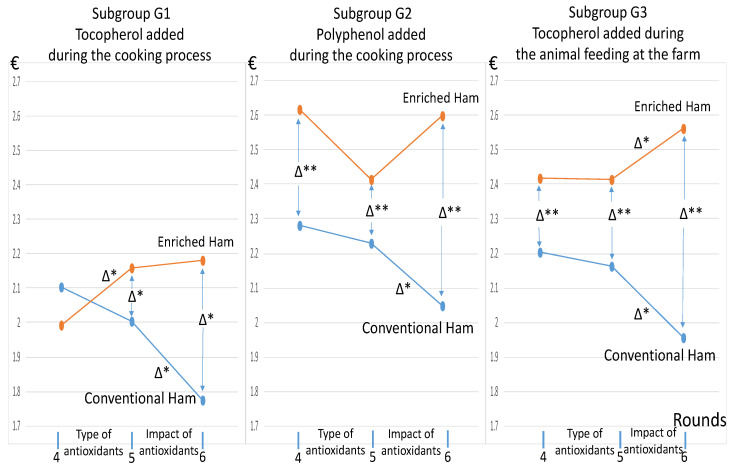
The average WTP for one packet of ham (€) and the influence of specific messages about antioxidants at round #5 for the subgroups G1, G2, and G3. Note: there were 44 participants for subgroup G1, 45 participants for subgroup G2, and 50 participants for subgroup G3. Δ* denotes a significant difference at 5%, and Δ** denotes a significant difference at 1%, as tested by the Wilcoxon test for comparing paired sample choices. See the note of Figure 1 for the interpretation of the tests.

**Figure 3 nutrients-13-01542-f003:**
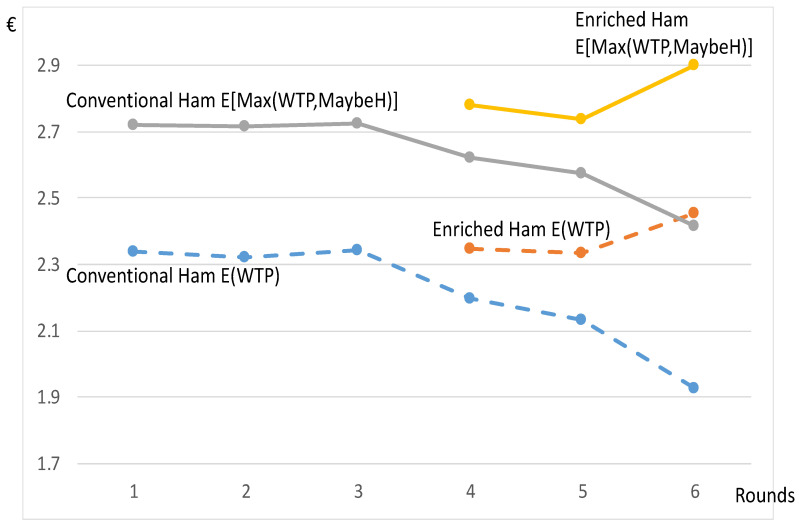
Average Max[WTP,MaybeH] and average WTP for Hams (€).

**Figure 4 nutrients-13-01542-f004:**
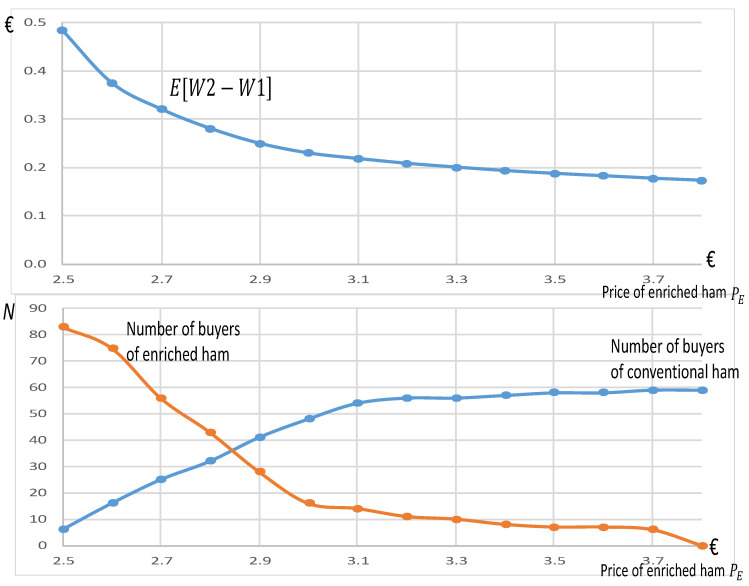
Average variations of consumers’ surpluses for one packet of enriched ham and the number of buyers following the introduction of enriched ham under perfect information.

**Figure 5 nutrients-13-01542-f005:**
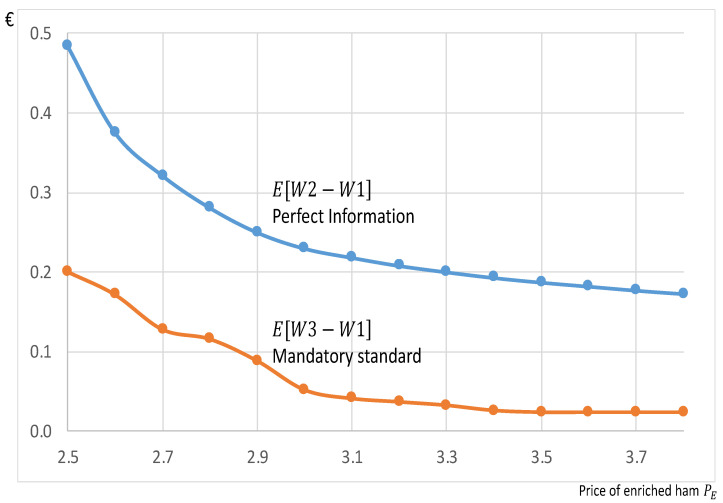
Average variations of consumers’ surpluses for one packet of enriched ham with a mandatory standard.

**Table 1 nutrients-13-01542-t001:** Sociodemographic characteristics of participants.

		This Experiment 139 Participants	French Population ^1^
**Gender**	Women (%)	51.8	51.6
	Men (%)	48.2	48.4
**Age (year)**	20–49 (%)	47.3	65.6
	50 and over (%)	52.7	34.4
**Level of education ^2^**	<Baccalaureate (%)	23.7	28.4
	Bac and bac + 2 (%)	46.1	40.3
	Higher than bac + 2 (%)	30.2	31.3

Note: ^1^ 2018 figures, INSEE (National Institute of Statistics and Economic Studies). ^2^ Baccalaureate (bac): French high school diploma.

**Table 2 nutrients-13-01542-t002:** The number of participants with specific reactions.

Consumers	Characterization	Number of Participants
Non-Engaged	WTPC#1 = WTPC#6 = WTPE#6 = 0	11
Indifferent	WTPC#6 = WTPE#6 > 0	21
Boycotter of enriched ham	WTPC#6 > 0 and WTPE#6 = 0	11
Reluctant	WTPC#6 > WTPE#6 > 0	1
Sensitive to messages	WTPE#6 > WTPC#6 > 0	69
Boycotter of conventional ham	WTPE#6 > 0 and WTPC#6 = 0	19
New Consumers	WTPC#1 = WTPC#6 = 0 and WTPE#6 > 0	7

Note: WTPC#*R* and WTPE#*R*, respectively, denote the WTP for conventional and enriched products at round #*R.*

**Table 3 nutrients-13-01542-t003:** Estimations of the pooled WTP for one packet of ham, either conventional or enriched with antioxidants, with a Tobit random effects estimator (rounds *R* = {4,5,6}).

Independent Variables	WTP Model 1	WTP Model 2
Conventional ^a^	2.713 ** (0.089)	2.717 ** (0.088)
Enriched ^b^	2.870 ** (0.089)	2.704 ** (0.105)
Enriched ^b^ × (age > 50 ^c^)		0.312 ** (0.106)
Conventional ^a^ × (message on antioxidants in round #5 ^d^)	−0.081 (0.118)	−0.082 (0.117)
Enriched ^b^ × (message on antioxidants in round #5 ^d^)	−0.019 (0.117)	−0.018 (0.117)
Conventional ^a^ × (scientific message on health impacts in round #6 ^e^)	−0.341 ** (0.119)	−0.340 ** (0.118)
Enriched ^b^ × (scientific message on health impacts in round #6 ^e^)	0.120 (0.117)	0.121 (0.116)
Stand. devi ε ^f^	0.950 ** (0.027)	0.944 ** (0.027)
Stand. dev. µ ^g^	1.296 ** (0.056)	1.301 ** (0.055)
Observations	*N* = 834	*N* = 834
Log likelihood	−1197.56	−1193.33

Note: ** significant at 1%. Standard errors in parentheses; ^a^ 1 if conventional ham, 0 otherwise; ^b^ 1 if enriched ham, 0 otherwise; ^c^ 1 if age of participant > 50, 0 otherwise; ^d^ 1 if WTP at round#5, 0 otherwise; ^e^ 1 if WTP at round#6, 0 otherwise; ^f^ standard deviation related to the random effect of the estimator; ^g^ standard deviation related to the Tobit part of the estimator.

**Table 4 nutrients-13-01542-t004:** Opinions about the main labels posted on ham packets sold in France. Labels and allegations presented to participants with their logo in the exit questionnaire (% of respondents).

	Organic	Label Rouge	GI ^c^	French Origin	BleuBlanc Coeur	Nutri-Score	DoNot Know
**Labels**	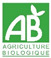	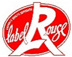	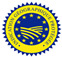	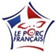	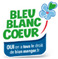	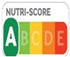	
The best quality ^a^	19.6%	37.0%	1.4%	10.1%	5.8%	15.2%	10.9%
The most useful for purchasing ^b^	21.7%	23.9%	3.6%	13.7%	4.4%	23.1%	9.4%
**Allegations**	**Without** **nitrite**	**−25% of salt**	**Without** **antibiotic**	**Do not** **know**		
The best quality ^b^	26.8%	9.4%	50.7%	13.1%		
The most useful for purchasing ^c^	29.7%	11.6%	49.3%	9.4%		

Note: ^a^ From the exit questionnaire, the exact question was “Which label (or allegation) is the most efficient in ensuring a good quality of the meat?” ^b^ The exact question was “Which label (or allegation) seems the most useful for guiding your future purchases of ham?” ^c^ GI means geographic indication, and the European label was shown.

**Table 5 nutrients-13-01542-t005:** Adhesion to colorectal cancer screening: % of “yes” answers among respondents.

	Age < 50	Age ≥ 50
Screening test realized over the last two years ^a^		52
Reinforced wish for a future screening test ^b^	79.6	82.6
Participants ready to pay €10 for a test ^c^	92	90.6
Participants ready to pay €20 for a test ^c^	77.7	70.6

Note: ^a^ From the exit questionnaire, the exact question for only participants older than 50 was “Over the last two years, did you realize a screening test for preventing a colorectal cancer?” ^b^ The exact question was “Did the revealed information reinforce your wish for realizing a future screening test?” with the additional text “when you will be 50” for participants younger than 50. ^c^ The exact question was “If the test was not reimbursed by the national medical insurance (Sécurité Sociale), would you be ready to pay X € for realizing this test?” with X = {€10, €20}.

## Data Availability

Data may be provided upon request.

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
