# Peer review of "Regulation and Consumer Interest in an Antioxidant-Enriched Ham Associated with Reduced Colorectal Cancer Risks"

_nutrients, 2021, doi:10.3390/nu13051542_

Round 1
Reviewer 1 Report
The present manuscript entitled "Regulation and consumer interest in an antioxidant-enriched ham associated with reduced colorectal cancer risks" is an article that tries to evaluate the attitudes of consumers towards two ham products associated with different risks of colorectal cancer. The article is detailed below point by point:
1. The last paragraph of the introduction is not an introduction, it does not contribute anything to the theoretical framework of the introduction (it is not possible to detail where it appears since the lines of the article are not numbered).
2. The background section is not in accordance with the regulations of the magazine.
3. The citations in the text show how the authors have not reviewed the regulations of the journal or the article before sending it, since they mix APA style with Vancouver citation.
4. Figure 1 does not contribute anything to the study, apart from being difficult to read.
5. After the fifth paragraph of the discussion, it is no longer a discussion, since it is not compared with other studies, it is to extend the results.
6. The conclusion is clear and ends with the implication to the practice of the subject of study.
7. Bibliographic references do not comply with the journal's regulations.
A study that is to be published in a Q1 cannot have so many errors and there is the present lack of review by the authors.
Author Response
Response to Reviewer #1 Comments
We thank the reviewer #1 for her/his thoughtful comments. We did our best to incorporate all of them into the revised version. We are grateful to the reviewer for his insights and we feel that the revised version significantly improved over the original.
All our modifications are indicated in red in the new draft. The few crossed-out parts in red will disappear in the next draft.
Thanks again!
Please find our reply to your comments!
- The last paragraph of the introduction is not an introduction, it does not contribute anything to the theoretical framework of the introduction (it is not possible to detail where it appears since the lines of the article are not numbered).
We agreed and we withdrew this last paragraph (see the cross-out part).
The background section is not in accordance with the regulations of the magazine.
We agreed and we merge the background section in the new introduction.
- The citations in the text show how the authors have not reviewed the regulations of the journal or the article before sending it, since they mix APA style with Vancouver citation.
We agreed and complied with the APA style with Vancouver citation.
- Figure 1 does not contribute anything to the study, apart from being difficult to read.
We agreed and got rid of Figure 1 (namely the picture of the package in the previous picture). The picture is now in the Appendix A of the supplementary material.
After the fifth paragraph of the discussion, it is no longer a discussion, since it is not compared with other studies, it is to extend the results.
The sections 4 is now divided in two subsections, namely the discussion (4.1) and the policy implication in subsection (4.2).
The conclusion is clear and ends with the implication to the practice of the subject of study.
Thanks for this comment.
- Bibliographic references do not comply with the journal's regulations.
We now comply with the journal’s regulation.
Thank you again for these precious comments.
Reviewer 2 Report
nutrients-1184646-peer-review-v1-rev
The manuscript entitled "Regulation and consumer interest in an antioxidant-enriched ham associated with reduced colorectal cancer risks" addresses a very current and highly relevant topic in the sector of processed pork products, such as ham. The manuscript is based exclusively on non-experimental work, but on the results of a survey carried out in France, where respondents are asked about their decision to purchase based on a hypothetical (not yet commercialized) ham product enriched with antioxidants, compared to conventional ham, without antioxidants.
I am of the opinion that the manuscript may be published after the introduction of several corrections, which I suggest below.
Corrections needed: The authors did not respect the rules in force in the MDPI journals, since the citation of bibliographic references should be made by numbers between [] and not by (author year). They will have to make this correction throughout the text
Corrections to be made at the structure level of the entire manuscript:
pg. 4 - Please confirm the meaning of the phrase: "to recruit half of participants younger than 50 and half of participants older than 50. Participants were recruited by email. They were informed that the experiment would focus on pork consumption and it would last about one hour with a 10 € participation fee. Only participants, eating pork or processed meat even occasionally, were selected. Each experimental session lasted 50 minutes in average and included between 14 and 18 participants."
pg. 5 - With pictures of products shown on Figure 1, people were asked to indicate choices they would make in supermarkets.
pg. 5 - The experiment focused on a 160 g package of ham containing 4 slices of ham.
pg. 5 - The observed market prices ranged from 2.4 to 3.2 € at the time preceding the experiment.
pg. 6 - Thus, the PNNS (French National Health Nutrition Program) recommends limiting the consumption of charcuterie products to 150 g per week.
pg. 6 - Epidemiological studies have shown that people who consume more than 50 g/day of processed meats, see the risk of colorectal cancer (concerning the colon and rectum) increase by 16%.
pg. 6 - However, a reasonable consumption of meat and processed meats is of nutritional interest with the provision of iron (very absorbable heme iron and important to fight against anemia), vitamin B12 (essential for neurological function and meat products are the main source), and protein of good nutritional quality.
pg. 6 - to limit the consumption of red meats to 500 g per week and the consumption of processed meats to 150 g per week, and among the processed meat to favor white ham and poultry ham,
pg. 7 - During each round, participants were asked to choose whether or not they will buy the product for prices varying from 2.20 to 3.80 € for conventional ham (see Appendix A).
pg. 8 - If one participant only replied “no” or “maybe” to each line, the selected WTP was equal to 0 € (the alternative configuration with a value equal to 2.10 € was also studied). If one participant only replied “yes” to each line, the selected WTP was equal to 3.80 €.
pg. 9 - The impact of revealed messages on average WTP for one ham packet of 160 g is shown in Figure 2, taking into account all WTP, including the non-engaged bidders with WTP equal to zero (explaining why averages may be lower than 2.2 €, the lower bound of the multiple-price list).
pg. 9 - The rounds R={1,…,6} are represented on the X-axis with indications on the types of information, and the WTP in € (euros) on the Y-axis.
pg. 9 - In Figure 2, the indicators Δ isolate the significant impact of a single round of additional information for a same ham, and/or between the two types of hams for a same round.
pg. 10 - Eventually, Figure 2 shows that both WTP for hams are impacted by the last messages about the reduction in the risk of colorectal cancer coming from the enriched ham.
pg. 12 - The impact of differentiated messages on the types of antioxidants at round #5 is now presented. Figure 3 is close to Figure 2, except that we detail the subgroups and we restrict our attention to rounds R={4,5,6} with the presence of both hams.
pg. 12 - Table 4 confirms results of the right part of Figure 2 and underlines a significant...
pg. 15 - of 139 participants, in particular for values between 2.5 and 2.9 €.
pg. 16 - ...last Table 6 focuses on questions related to the participants’ intentions regarding colorectal cancer screening aiming at reducing risks of cancer (Frew et al., 2001).
Table 6 - Ready to pay 10 €
Ready to pay €20 €
pg. 16 - ...would you ready to pay X € for realizing this test?” with X = {10 €, 20 €}.
pg. 18 - per package for a price (??) ≤ 2.80 € (Figure 6).
pg. 18 - ... price would be equal to ?? ≤ 2.516 €, leading to a welfare estimate of ?[?3 − ?1] = 0.19 € per package
pg. 18 - Eventually, we extrapolate this average surplus variation for 160 g package of pork,
pg. 18 - 6. Conclusions
pg. 19 - 7. References
Author Response
Response to Reviewer# 2 Comments
We thank the reviewer #2 for her/his thoughtful comments. We did our best to incorporate all of them into the revised version. We are grateful to the reviewer for his insights and we feel that the revised version significantly improved over the original.
All our modifications are indicated in red in the new draft. The few crossed-out parts in red will disappear in the next draft.
Thanks again!
Please find our reply to your comments!
Corrections needed: The authors did not respect the rules in force in the MDPI journals, since the citation of bibliographic references should be made by numbers between [] and not by (author year). They will have to make this correction throughout the text
We agreed and complied with the APA style with Vancouver citation. We now follow the guideline of the journal!
Regarding all the following comments, we agreed and directly improved the underlined parts directly in the draft. Please refer to the modifications in red in the draft.
Thanks again for your precious comments.
Round 2
Reviewer 1 Report
After the authors' revisions, I accept the manuscript.
Reviewer 2 Report
The authors proceeded to a profound alteration of the manuscript in accordance with the reviewers' comments, so it is much better than the original version.